# Transcriptomic Analysis of Drug-Resistance *Acinetobacter baumannii* under the Stress Condition Caused by *Litsea cubeba* L. Essential Oil via RNA Sequencing

**DOI:** 10.3390/genes12071003

**Published:** 2021-06-29

**Authors:** Yunqiao Yang, Kaiyuan Hao, Mingsheng Jiang, Fareed Uddin Memon, Lei Guo, Geyin Zhang, Tian Liu, Xianshi Wu, Hongbin Si

**Affiliations:** 1College of Animal Sciences and Technology, Guangxi University, Nanning 530004, China; yyqop01@163.com (Y.Y.); 1918302008@st.gxu.edu.cn (K.H.); 13878896387@163.com (M.J.); fareedjanm@gmail.com (F.U.M.); 18838516358@163.com (G.Z.); 1834130313@st.gxu.edu.cn (T.L.); xianshiw@gxu.edu.cn (X.W.); 2College of Bioscience and Biotechnology, Yangzhou University, Yangzhou 225012, China; 007005@yzu.edu.cn

**Keywords:** *Litsea cubeba* L., plant essential oil, *Acinetobacter baumannii*, biofilm, RNA sequencing

## Abstract

*Litsea cubeba* L. essential oil (LCEO) can affect the growth of drug-resistance bacteria. However, research on stress response of drug-resistant *A. baumannii* under sub-lethal LCEO concentrations had been limited so far. Therefore, transcriptomic analysis of *A. baumannii* under 1/2 minimum inhibitory concentration (MIC, 0.54 mg/mL) of LCEO was performed. Results of transcriptomic analysis showed that 320/352 genes were significantly up/down-regulated, respectively, in LCEO-treated *A. baumannii*. Both up and down-regulated genes were significantly enriched in three GO terms (oxidation-reduction process; oxidoreductase activity; oxidoreductase activity, acting on the CH-CH group of donors), which indicated that the redox state of *A. baumannii* was significantly affected by LCEO. LCEO may also inhibit aerobic respiration, synthesis of ketone bodies and the metabolism of some amino acids while, meanwhile, promoting fatty acid degradation of *A. baumannii* according to Kyoto Encyclopedia of Genes and Genomes (KEGG) enrichment. The permeability and the stress of cell membrane of *A. baumannii* were significantly affected by LCEO. After crystal violet dyeing, the biofilm formation of *A. baumannii* was promoted/inhibited by extremely low/relatively high concentration of LCEO, respectively. LCEO and chloramphenicol have synergistic growth inhibitory effect against *A. baumannii* according to the Fractional Inhibitory Concentration Index (FICI) value = 0.375. Our results indicate that the growth of *A. baumannii* was inhibited by LCEO, and give insights into the stress response of *A. baumannii* under sub-lethal concentrations of LCEO. These results provided evidence that *A. baumannii* was inhibited by LCEO, and expanded knowledges of stress response of *A. baumannii* under sub-lethal concentration of LCEO.

## 1. Introduction

LC is a *Lauraceae* is a Lauraceae plant and widely grown in East Asia [1]. The whole plant of *Litsea cubeba* L. gives off a smell that is similar to ginger [2]. According to the records of pharmacology and herbal medicine, *L. cubeba* has therapeutic effects on skin, lung and breast infections [3,4]. Research reports on pharmacology and chemical composition of *L. cubeba* essential oil (LCEO) have gradually increased in recent years. Ho’s results showed that limonene, citronellal, and citronellol were the main components of LCEO in the stem of *L. cubeba*. The content of citral, 1,8-cineole was higher in the parts where the content of water is higher, such as flowers, twig and leaves [2]. The fruit oil of LC can be used as a flavor enhancer for cigarettes, cosmetics and food products, as well as as a food preservative at present [1,5]. In the process of industrial production of neral and geranial, LCEO were often used as raw material [6]. The antioxidative [7,8], anticancer insecticidal [9] and antimicrobial activities [1,10] of LCEO were reported in pharmacological research fields. Studies have shown that LCEO can affect the cell membrane stability of Gram-negative and Gram-positive bacteria, which leads to the death of the cells, and can also deal with drug-resistance bacteria [7,11,12,13].

*A. baumannii* is a successful pathogen which often causes infections of humans and aquatic animals [14]. It usually causes bacteremia, pneumonia, meningitis, peritonitis, endocarditis, and urinary tract and skin infections [15,16].

However, few antibiotics can cure infections and prevent outbreaks caused by *A. baumannii*, as this pathogen has multi drug-resistant properties [17]. To address drug-resistant infections, Adukwu indicated the potential use of lemongrass (*Cymbopogon flexuosus*) essential oil (LEO) and citral which have the ability to inhibit and kill multi drug-resistant (MDR) *A. baumannii* [18]. The main component of LCEO is citral, however, the impact of LCEO on *A. baumannii* has not been reported yet. Therefore, the elucidation of the mechanisms by which LCEO impacts *A. baumannii* would be helpful for the development and clinical utilization of LCEO.

In our previous work, through transcriptomics, we found that toxin production and a variety of metabolic pathways of *Staphylococcus aureus* (*S. aureus*) were significantly affected by LCEO [19]. The objective of the present work was to detect stress reactions of MDR *A. baumannii* under 1/2 minimum inhibitory concentration (MIC) of LCEO using transcriptomics.

## 2. Materials and Methods

### 2.1. Chemical and Bacterial Strains

LCEO was provided by Jiangxi Hongwei Natural Perfume Oil Co. LTD (China). As shown in Table 1, the chemical compositions of LCEO in this work were listed. We mixed 14.29 wt% LCEO, 14.29 wt% tween-80 and 71.43 wt% distilled water to form a stable system which can disperse the LCEO in water. In this system, the hydrophilic-lipophilic balance value was 15.

Three strains of *A. baumannii* (*A. baumannii* 4, *A. baumannii* 17 and *A. baumannii* 20) were provided by the First Affiliated Hospital of Guangxi Medical University, China. These three strains were all harboring *arm-A* (β-lactam resistance genes), *sul2* (sulfa resistance genes), *tetB* (tetracycline resistance genes). *A. baumannii* 17 also harbors *aac6′-1b* (aminoglycoside resistance genes). In addition, *A. baumannii* 17 has a strong virulence: when 2 × 10^8^ colony-forming units (CFU)/mL of this fresh strain was intraperitoneally injected at a dosage of 0.1 mL/10 g body weight for a mouse, the mortality rate was 100% within 12 hours (12 of 12 mice).

For each strains, 0.5 mL of bacteria was taken from the cryotube and then added to a tube with 3.5 mL tryptic soy broth (TSB) for 24 h shaking at 37 ℃, at 300 RPM. The culture medium was inoculated on tryptone soy agar (TSA) plates using an inoculation ring. After these plates were cultured for 24 h at 37 °C, a single colony was selected for inoculation to TSB culture for 6 h at 37 °C at 300 RPM shaking before subsequent trials.

### 2.2. Determination of Minimum Inhibitory Concentration (MIC)

Serial-dilution culture method was performed to obtain MIC values for different antibiotics (ceftriaxone sodium, amoxicillin, berberine, neomycin, levofloxacin, fosfomycin, colistin, mequindox, clinfloxacin, sulfamonomethoxine, gatifloxacin, amikacin, ceftazidime, lincomycin, ceftiofur sodium, florfenicol, azithromycin, cefotaxime sodium, rifampin, meropenem, ceftriaxone sodium), tween-80 and LCEO on *A. baumannii*.

Every drug solution was two-fold diluted in TSB and equally mixed with 100 μL *A. baumannii* 17 TSB suspensions (10^4^ CFU/mL) respectively in a 96-well microtiter plate. After stationary culture at 37 °C for 24 h, MIC values were judged by no visible growth of *A. baumannii* of broth in wells [20]. This test was repeated three times.

### 2.3. Growth Curve

A growth curve assay was performed to investigate the growth inhibitory effect of LCEO on *A. baumannii* 17. Bacterial TSB suspensions (10^4^ CFU/mL) were transferred into tubes, followed by tween 80 and LCEO. The final concentration of tween 80, LCEO (1/2 MIC) and LCEO (1 MIC) in corresponding groups was 1.08, 0.54 and 1.08 mg/mL. After 300 RPM shaking at 37 °C for 24 h, the OD_600_ was determined at intervals by a spectrophotometer. Before each detection, zero was set using a same batch of sterile TSB. This test was repeated three times.

### 2.4. Time Kill Curve

After 12 h culture at 37 °C with 300 RPM shaking, *A. baumannii* 17 culture was added with tween 80 and LCEO. The final concentration of tween 80, LCEO (1/2 MIC) and LCEO (1 MIC) in the corresponding group was 1.08 mg/mL, 0.54 and 1.08 mg/mL. Next, all samples were cultured for another 8 h. At intervals, 100 μL culture in each sample was taken, centrifuged and resuspended in 100 μL sterile phosphate-buffered saline (PBS). Subsequently, 10-fold serially diluted suspensions were plated on TSA plates and incubated 24 h at 37 °C. Colonies were counted and the CFUs/mL was calculated. This experiment was performed with at least three biological replicates.

### 2.5. Preparation of Bacterial Samples for RNA-Seq

For RNA extraction, culture of *A. baumannii* 17 was inoculated into fresh TSB then incubated at 37 ℃ with 250 RPM shaking. When the opioid addiction (OD) value at 600 nm of culture reached 0.8 (1 × 10^6^ CFU/mL), ½ MIC (0.54 mg/mL) of LCEO was set as stimulation treatment, thus bacteria from the aliquot were suspended in 10 mL TSB with ½ MIC LCEO, and the bacterial from another aliquot were suspended in 10 mL TSB as control.

When the OD_600nm_ value of the culture reached 0.8 (1 × 10^6^ CFU/mL), ½ MIC of LCEO was added in the culture, and this was the LCEO treatment group. At the same time, a control group was set without LECO. Each group had three independent samples. After 30 min incubation at 37 °C, bacterial cells were collected for RNA extraction.

### 2.6. RNA Extraction and Purification, cDNA Library Construction for RNA-Seq, RNA-Seq and Data Analysis

By using the Total RNA Isolation System (Promega), total RNA of six cell samples was extracted. The extracted RNAs quality was checked using Agilent 2100 Bioanalyzer. Library construction and Illumina sequencing was undertaken by Novogene China.

All RNA-seq reads were aligned to the genome of *A. baumannii* ATCC 17978 (https://www.ncbi.nlm.nih.gov/assembly/GCA_000015425.1; accessed on 1 March 2007) for data analysis. Only genes with an adjusted *p*-value  <  0.05 and |fold change|  >  1.5 were regarded as differentially expressed genes (DEGs).

### 2.7. Leakages of Intracellular DNA and Protein

To investigate the leakages of intracellular DNA and proteins of *A. baumannii* 17 triggered by LCEO, an ultraviolet (UV) light spectrophotometer (DU 730 Nucleic Acid/Protein Analyzer; Beckman Coulter, Brea, CA, USA) was used [21]. A fresh culture of *A. baumannii* 17 (1 × 10^6^ CFU/mL, TSB cultured) was washed with sterile water twice and then resuspended by sterile water to the same volume. The resuspended *A. baumanni* from the previous step was divided into 16 test tubes (4 groups and 4 duplicates). The four groups were: Control, 1/8 MIC (0.135 mg/mL of LCEO), 1/4 MIC (0.27 mg/mL of LCEO) and 1/2 MIC (0.54 mg/mL of LCEO). These 16 test tubes were incubated at 37 °C with 250 rpm shaking for 2 h. Afterwards, 16 test tubes were subjected to centrifugation under 5500 rpm for 15 min, and then the supernatant was taken for absorbance detection of OD_260nm_ (DNA absorbance) and OD_280nm_ (protein absorbance). Because citral (the main active component of LCEO) absorbs UV light, three control tubes with the same concentration of LCEO were set for zeroing the OD value before testing, respectively.

### 2.8. L. cubeba Essential Oil (LCEO) on Biofilm Formation

The method applied was a slight modification of the method used in a previous work [22]. LCEO (from 0.54 to 0.00026 mg/mL and the 0 mg/mL was set as control) was two-fold diluted in TSB and equally mixed with 100 μL *A. baumannii* TSB suspensions (10^4^ CFU/mL) respectively in a 96-well microtiter plate. After incubation at 37 °C for 36 h, the culture was removed and 200 μL methanol was added to each well for 15 min at room temperature. After that, the methanol supernatant was removed, and the plates were dried at room temperature. Then, 200 μL of 0.1% crystal violet solution was added to the wells for 15 min at room temperature. After that, the crystal violet was removed, the wells were washed gently with distilled water until the distilled water was clear and transparent, and 200 μL 33% acetic acid was added to each well. Finally, the OD_450nm_ value of plates were read [22]. This test was repeated at three times.

### 2.9. Fractional Inhibitory Concentration Index (FICI)

The Fractional Inhibitory Concentration Index (FICI) was used to evaluate the synergistic growth inhibitory effect of LCEO with other antibiotic (tetracycline, meropenem, chloramphenicol, amikacin, ampicillin, kanamycin, ciprofloxacin, fosfomycin, streptomycin sulfate, ceftiofur). FICI = FICA + FICB = C _A_^comb^/MIC _A_ ^alone^ + C _B_ ^comb^/MIC _B_ ^alone^. MIC _A_ ^alone^ and MIC_B_ ^alone^ are the MIC values of drug A and B when acting alone. C _A_^comb^ and C _B_ ^comb^ are concentrations of drug A and B in combination. FICI results were determined as follows: synergy: <0.5; additivity: 0.5–1; indifference: >1–4; and antagonism: >4 [23,24]. This test was repeated at three times.

### 2.10. Quantitative Real-Time Polymerase Chain Reaction (RT-PCR)

To validate the RNA-seq results, 8 DEGs were selected for quantitative real-time PCR (RT-PCR) assay. The selective criteria of these 8 DEGs were: (1) in different Kyoto Encyclopedia of Genes and Genomes (KEGG) pathways; (2) the |fold change| > 1.5; (3) 4 DEGs were up-regulated and the remaining 4 DEGs were down-regulated. These experiments were performed in triplicate.

Total RNA was extracted from the bacteria using Bacteria RNA Plus Reagent (Vazyme) and RNA Isolater Total RNA Extraction Reagent (Vazyme). HiScript^®^ III RT SuperMix for qPCR (+gDNA wiper) (Vazyme) was used for reverse transcription. Primers were designed by Primer-BLAST (NCBI) and provided by Vazyme (Table 2). The amplification and detection part were carried out by LightCycler^®^ 480 (Roche). This test was repeated three times.

### 2.11. Statistical Analysis of Quantitative Real-Time PCR and Opioid Addiction (OD) Value

The statistical analysis of results was performed using one-way analysis of variance (ANOVA) followed by post hoc analysis with Tukey’s test using SPSS16 software. Values of *p* < 0.05 were considered to be statistically significant.

## 3. Results and Discussion

### 3.1. MIC Values

All strains had obvious drug-resistant phenotype to some common antibiotics: amoxicillin, amikacin, sulfamonomethoxine, azithromycin (Table 3). However, the MIC value of tween 80 on *A. baumannii* was >108 mg/mL, which may indicate that the tween 80 has no negative effect on the growth of *A. baumannii*.

We selected *A. baumannii* 17 for follow-up studies because the *A. baumannii* 17 harbored more drug resistance gene than other 2 strains, and *A. baumannii* 17 also had strong virulence phenotypes as mentioned above.

### 3.2. Growth Curve and Time Kill Curve

As shown in Figure 1A, the growth of *A. baumannii* 17 was inhibited by LCEO. However, tween 80 had no such effect. Similar results were found in the time kill assay (Figure 1B). Combined with the above MIC results and the results of existing studies [25,26], it was easy to assume that in this LCEO dispersion system, only LCEO plays a bacteriostatic role.

### 3.3. General Transcriptome Information

RNA-Seq generated total 10,597,964 to 20,954,540 clean reads from LCEO-treated (AB_LCEO_treated) and control cDNA libraries, respectively (Appendix A). These clean reads were mapped to the reference genome of *Acinetobacter baumannii ATCC 17978* (https://www.ncbi.nlm.nih.gov/assembly/GCA_000015425.1; accessed on 1 March 2007).

As the gene expression profile of samples in the same group were highly similar, further analysis can be performed. Per Kilobase of transcript per Million mapped reads (RPKM, full information of reads quantification) analysis showed that 4320 total genes were expressed in LCEO-treated and non-treated groups.

The raw sequence data reported in this paper have been deposited in the Genome Sequence Archive (Genomics, Proteomics and Bioinformatics 2017) in BIG Data Center (Nucleic Acids Res 2019), Beijing Institute of Genomics (BIG), Chinese Academy of Sciences, under accession numbers CRA003099 that is accessible at https://bigd.big.ac.cn/gsa (accessed on 18 August 2020).

### 3.4. Differentially Expressed Genes in LCEO-Treated A. Baumannii 17

To understand and explain the results of RNA-seq, the differential expression patterns of the transcripts were analyzed between the LCEO-treated and control group. 672 genes (320 up and 352 down-regulated) were identified by differential expression analysis (Figure 2). Among these 672 DEGs, the |fold change| of 11 up and 9 down-regulated DEGs were all >4.

According to the gene ID and gene name of the NCBI database, 11 up-regulated genes encode 4-carboxymuconolactone decarboxylase; NAD(P)-dependent alcohol dehydrogenase; alkane 1-monooxygenase; AraC family transcriptional regulator; hypothetical protein; N/A; 3-oxoacid CoA-transferase subunit A; hypothetical protein; α/β fold hydrolase; L-lactate permease; chromate transporter. The 9 down-regulated genes encode ferredoxin reductase; acyl-CoA desaturase; VOC family protein; 4-hydroxyphenylpyruvate dioxygenase; maleylacetoacetate isomerase; amino acid permease, fumarylacetoacetase; 1,2-phenylacetyl-CoA epoxidase subunit B; SDR family oxidoreductase.

### 3.5. Gene Ontology (GO) Functional Enrichment Analysis

Gene Ontology (GO) is an international standard classification system of gene function. By GO enrichment analysis of mRNA, functions of targeted mRNA can be obtained.

All 672 DEGs underwent GO enrichment analysis; 848, 183 and 529 specific GO terms in biological process, cellular component and molecular function were reported, as shown in Figure 3 and Appendix A.

Ten GO terms were enriched in biological process; including oxidation-reduction process; anion transport; organic anion transport; organic acid transport; carboxylic acid transport; anion transmembrane transport; organic acid transmembrane transport; amino acid transport; ion transmembrane transport; single-organism metabolic process.

Ten GO terms were enriched in cellular components; including membrane-enclosed lumen; organelle lumen; intracellular organelle lumen; nuclear part; nuclear lumen; proton-transporting two-sector ATPase complex, catalytic domain; intracellular organelle part; THO complex; chromosome; dynein complex.

In the category of molecular function, 10 GO terms (oxidoreductase activity; acyl-CoA dehydrogenase activity; coenzyme binding; oxidoreductase activity, acting on the CH-CH group of donors; cofactor binding; flavin adenine dinucleotide binding; excitatory extracellular ligand-gated ion channel activity; extracellular-glutamate-gated ion channel activity; organic acid transmembrane transporter activity; carboxylic acid transmembrane transporter activity) were significantly enriched.

Within the 30 significantly enriched GO terms, there were six terms containing the keyword “membrane”. Combined with a recent report [27], LCEO have significant negative effects on the cell membrane and cell wall of *A. baumannii* 17.

Previous studies have shown that LCEO has a strong antioxidant activity (that is, reducing activity) [7,28]. It was noteworthy that in a total of 672 DEGs, 127 DEGs were involved in three GO terms which contained the prefix “oxid”. The three GO terms were: oxidation-reduction process; oxidoreductase activity; oxidoreductase activity, acting on the CH-CH group of donors. These results suggested that LCEO influenced the biological redox processes of *A. baumannii* 17 comprehensively. We wanted to know which pathways that these 127 DEGs were involved in. Therefore, a customized KEGG enrichment analysis was performed individually based on the 127 DEGs.

### 3.6. Kyoto Encyclopedia of Genes and Genomes (KEGG) Pathway Enrichment Analysis

KEGG is a database integrating genome, chemical and system functional information. KEGG pathway enrichment analysis can not only find out the pathways involved in target genes or proteins, but also analyze the significant differences of these pathways.

To further identify significant changes in the biochemical pathways during LCEO stimulation, the 672 DEGs were further mapped using the KEGG database (Appendix A). In Figure 4, 12 KEGG pathways (valine, leucine and isoleucine degradation; propanoate metabolism; butanoate metabolism; synthesis and degradation of ketone bodies; fatty acid metabolism; fatty acid degradation; carbon metabolism; lysine degradation; geraniol degradation; phenylalanine metabolism; pyruvate metabolism; tryptophan metabolism) were significantly enriched.

Also, the 127 DEGs were mapped to the KEGG database (the reason was mentioned at the end of Section 3.5). Eight KEGG pathways (geraniol degradation; valine, leucine and isoleucine degradation; butanoate metabolism; fatty acid degradation; β-alanine metabolism; chloroalkane and chloroalkene degradation; lysine degradation; phenylalanine metabolism) were significantly enriched (Appendix A).

After a detailed comparison between the results of two KEGG enrichment analyses with 672 and 127 DEGs, we found that: the significantly enriched pathways were similar; the 127 DEGs were widely distributed in multiple KEGG pathways. Also, in the results of 127 DEGs, and in a particular KEGG pathway, the ratio of measured genes (input number) to annotated genes in the database (background number) was not high. These suggested that the biological redox processes of *A. baumannii* 17 were extensively and strongly interfered with by LCEO, and these effects may be non-specific. Therefore, we hypothesized that many redox reactions in cells were affected by LCEO as LCEO can simultaneously act on multiple targets of *A. baumannii* 17.

In the KEGG enrichment sections below, only the results of the total 672 DEGs were discussed in detail.

### 3.7. Differentially Expressed Genes (DEGs) Involved in “Citrate Cycle (TCA Cycle)” Pathway

It was necessary to discuss the citrate cycle (TCA cycle) pathway first, although it was not significantly enriched (*p* = 0.0780). TCA cycle is an important aerobic pathway for the final oxidization of carbohydrates, fatty acids, proteins and other organics [29,30,31]. As compared to the control, in LCEO-treated *A. baumannii* 17, all the enriched genes were significantly down-regulated (Figure 5). The gene expressions of the key enzymes (citrate synthase [EC:2.3.3.1], isocitrate dehydrogenase [EC:1.1.1.42]) which were involved in two irreversible and critical rate-limiting reactions, was down-regulated. These results suggest that the declined reaction rate of the TCA cycle pathway, and a decreased energy production of the bacteria. The TCA cycle contains a number of dehydrogenation steps; however, the essence of dehydrogenation is oxidation. Then it made sense in principle that the TCA cycle of *A. baumannii* 17 was inhibited by LCEO with strong reducibility.

### 3.8. DEGs Involved in “Two-Component System”, “Valine, Leucine and Isoleucine Degradation”, “Tyrosine Metabolism” and “Phenylalanine Metabolism” Pathways

Two-component signal transduction systems enable bacteria to sense the changes and stimulation of the external environment, and carry out signal transmission. In general, the two-component system includes a membrane-bound histidine kinase and a corresponding response regulator. After the histidine kinase senses specific stimuli, it phosphorylates specific histidine residue of its own. Next, the corresponding response regulator mediates the cellular response. Histidine kinase and reaction regulator are the largest gene families in Gram-negative bacteria and cyanobacteria in particular [32,33,34,35]. AauSR is a sensor of glutamate. In LCEO-treated *A. baumannii* 17, the expressions of downstream genes (*aatJMPQ*, *ansA*) were significantly downregulated as compared with the control (Figure 6A). These transcriptomic results strongly suggested a weakening of acidic amino acid (aspartic acid and glutamic acid) uptake and metabolism in the LCEO treated cell.

Valine, leucine and isoleucine can be degraded into the TCA cycle; pyrimidine metabolism; terpenoid backbone biosynthesis and other pathways [36,37]. In the LCEO-treated *A. baumannii* 17, genes encoding the enzymes that start the whole reaction (branched-chain amino acid aminotransferase [EC:2.6.1.42] and dihydrolipoamide dehydrogenase [EC:1.8.1.4]) were significantly downregulated in comparison with the control (Figure 6B). These results suggested that the activity of Valine, leucine and isoleucine degradation pathway was inhibited.

In the KEGG database, there were still many non-annotated genes in the tyrosine metabolism pathway map. As shown in Figure 6C, the expressions of genes encoding enzymes (4-hydroxyphenylpyruvate dioxygenase [EC:1.13.11.27], maleylacetoacetate isomerase [EC:5.2.1.2], fumarylacetoacetase [EC:3.7.1.2] and succinate-semialdehyde dehydrogenase [EC:1.2.1.16]) of irreversible reaction were down-regulated by LCEO in *A. baumannii* 17. Therefore, we can infer from the results a reduced production of succinate and fumarate for TCA cycle in cells.

Phenylalanine can be metabolized into succinyl-CoA and acetyl-CoA for the TCA cycle [38]. According to Figure 6D, gene expressions of almost all the enzymes known to be involved in this process were downregulated by LCEO significantly. Therefore, we can assume that the production of succinyl-CoA and acetyl-CoA from phenylalanine was decreased by LCEO of *A. baumannii* 17 as compared to that of the control.

### 3.9. DEGs Involved in “Two-Component System”, “Pyruvate Metabolism”, “Glycolysis/Gluconeogenesis” and “Oxidative Phosphorylation” Pathways

Aerobic respiration is the process of synthesizing large amounts of ATP and it is usually defined by the degradation of glucose. This process has three stages: 1 glycolysis; 2 TCA cycle; 3 oxidative phosphorylation (electron transport chain) [39,40,41]. As shown in Figure 6A, *CydAB* were down-regulated by LCEO of *A. baumannii* 17, which indicated that the aerobic respiration activity was slowed.

Pyruvate dehydrogenase complex (PDHC) is a mitochondrial multienzyme complex composed and it holds an important position in pyruvate metabolism pathway. PDHC is A group of rate-limiting enzymes that catalyze irreversible oxidative decarboxylation of pyruvate into acetyl-coA [42,43,44]. PDHC is composed of six enzymes, among which E1, E2 and E3 are the three main subunits of it [45]. As shown in Figure 7A, genes encoding PDHC (E1 component α subunit [EC:1.2.4.1], pyruvate dehydrogenase E2 component (dihydrolipoamide acetyltransferase) [EC:2.3.1.12] and dihydrolipoamide dehydrogenase: subunit E3 [EC:1.8.1.4]) in LCEO-treated *A. baumannii* 17 were down-regulated compared with that of the control. Also, LCEO has been shown to inhibit the hexose monophophate pathway of *S. aureus* [11]. These results suggested that LCEO had negative effects on mitochondrial energy metabolism in *A. baumannii* 17.

In the absence of oxygen, organisms derive energy from the glycolysis which is a preparatory pathway for aerobic oxidation of glucose in most organisms. Gluconeogenesis, in turn, is a synthesis pathway of glucose from non-carbohydrate precursors [46]. The gene expression of glucose-6-phosphate isomerase [EC:5.3.1.9] was down-regulated by LCEO. This enzyme ([EC:5.3.1.9]) catalyzes the reversible reaction among α-D-glucose 6-phosphate, β-D-fructose 6-phosphate and β-D-glucose 6-phosphate. At the same time, gene encoding aldose 1-epimerase [EC:5.1.3.3] that catalyzes the reversible reaction between α-D-glucose and β-D-glucose was down-regulated by LCEO (Figure 7B). These two enzymes are involved in both the glycolysis and gluconeogenesis pathways. Therefore; these results suggested that LCEO influenced the glycolysis/gluconeogenesis pathway of *A. baumannii* to some extent.

Oxidative phosphorylation is defined as an electron transfer chain driven by substrate oxidation that is coupled to the synthesis of ATP through an electrochemical transmembrane gradient in cytoplasm [47]. As shown in Figure 7C, all the enriched genes were down-regulated in known and relative independent steps. This suggested that the oxidative phosphorylation was also hindered.

We noticed that in Figure 6A, CydAB, PetABC and CycA/Y are controlled by RegAB and RegA is a sensor for sensing redox signals. The present data showed that both oxidative phosphorylation and aerobic respiration were inhibited. Therefore; it was reasonable to believe that LCEO may achieve these inhibitory effects through its reducibility via RegAB.

### 3.10. DEGs Involved in “Synthesis and Degradation of Ketone Bodies” Pathway

Acetyl-CoA, produced by the β-oxidation of fatty acids and other metabolisms, has another metabolic pathway. Finally, acetoacetic acid, -hydroxybutyric acid and acetone are formed. These three products are collectively called ketone bodies [48]. For bacteria, ketone bodies are a form of carbon and energy storage under nutrient deprivation, especially (R)-3-hydroxybutanoate [49]. However, excessive concentration of ketone bodies can inhibit the growth of bacteria [50]. Here, genes encoding hydroxymethylglutaryl-CoA lyase [EC:4.1.3.4] and 3-hydroxybutyrate dehydrogenase [EC:1.1.1.30] were down-regulated (Figure 8), which indicated that the reversible reaction between acetyl-CoA to (R)-3-hydroxybutanoate slowed down by LCEO. This may result in a decreased energy storage in the *A. baumannii* 17 treated with LCEO.

### 3.11. DEGs Involved in “Fatty Acid Degradation” and “Fatty Acid Biosynthesis” Pathways

The oxidative decomposition (degradation) of fatty acids is one of the important pathways for cells to obtain energy [51]. The fatty acids in cytoplasm are synthesized into hexadecanoyl-CoA by fadE then transported into the mitochondria via carnitine palmitoyl transferase I (CPT1) and CPT2. After seven cycles, hexadecanoyl-CoA is degraded into acetyl-CoA completely. There are four steps (dehydrogenation, hydration, dehydrogenation and transformation) in each cycle, and then a total of 4 mol ATP will be generated by the 2 dehydrogenation procedure in a cycle [51]. In current studies (Figure 9A), genes encoding enzymes (acyl-CoA dehydrogenase [EC:1.3.8.7], long-chain-acyl-CoA dehydrogenase [EC:1.3.8.8], enoyl-CoA hydratase [EC:4.2.1.17], 3-hydroxyacyl-CoA dehydrogenase [EC:1.1.1.35], acetyl-CoA acyltransferase [EC:2.3.1.16]) were up-regulated significantly, which suggested that the fatty acid degradation activity of *A. baumannii* 17 was increased by LCEO.

In the fatty acid biosynthesis pathway, hexadecanoyl-CoA is biosynthesized by acetyl-CoA by a series of reactions [52]. In Figure 9B, genes encoding proteins (acetyl-CoA carboxylase carboxyl transferase subunit α [EC:6.4.1.2], 3-oxoacyl-[acyl-carrier-protein] synthase I [FabB]) were down-regulated. This suggested that the fatty acid biosynthesis of *A. baumannii* 17 was slowed by LCEO. The degradation of fatty acid was enhanced while the synthesis of fatty acid was retarded, indicating that the cells were utilizing fatty acids to obtain energy.

Of course, some genes were down-regulated in the fatty acid degradation pathway; and some genes were up-regulated in the fatty acid biosynthesis pathway. However, we still believe that the overall trend was enhanced degradation and retarded biosynthesis of fatty acid. Because 1: in our previous work [19], LCEO caused a significant upregulation of genes related to fatty acid degradation in *S**. aureus*; 2: just a small proportion of genes were down-regulated in the fatty acid degradation pathway and up-regulated in the fatty acid biosynthesis pathway.

From this part, we assumed that LCEO suppressed the aerobic respiration and degradation of amino acid. At the same time, the concentration of ketone bodies used for energy storage was also decreased. In order to survive, *A. baumannii* 17 increased the activity of fatty acid degradation to obtain energy.

### 3.12. Cell Membrane Damage and DEGs Involved in “Two-Component System”, “Histidine Metabolism”, “Peptidoglycan Biosynthesis” Pathways

As shown in Figure 10A, the presence of LCEO significantly affected the permeability of cell membranes, which led to the leakage of proteins and nucleic acids. Studies have shown that LCEO is a cell wall-active agent and can cause cell membrane damage of *S. aureus* [11], and our current results were similar. *S. aureus* is a Gram-positive bacteria and *A. baumannii* 17 is a Gram-negative bacteria, which again showed that LCEO has spectral antimicrobial activity [7,53,54,55].

As shown in Figure 6A, the down-regulation of *EnvZ* (not shown in green frame as the |fold change| = 1.30) and *OmpR* suggests that the cells were stimulated by osmotic pressure (downshift of K^+^). This was probably due to damage to the cell membrane by the LCEO.

The importance and location of histidine kinases have been described above. Therefore, it is not difficult to speculate that histidine is particularly important in the repair process when the cell membrane is damaged. As shown in Figure 11A, Gene expression of all the annotated enzymes (histidine ammonia-lyase [EC:4.3.1.3], urocanate hydratase [EC:4.2.1.49], imidazolonepropionase [EC:3.5.2.7], formiminoglutamase [EC:3.5.3.8]) that catalyze the irreversible reactions of L-Histidine to L-Glutamate were significantly down-regulated. At the same time, the gene expression of the enzyme that catalyzes (histidinol dehydrogenase [EC:1.1.1.23]) the irreversible reaction (L-histidinol → L-histidinal → L-histidine) in the final step of histidine-synthesis was slightly up-regulated (not shown in green frame as the |fold change| = 1.28). According to the results of cell contents leakage and RNA-Seq, we hypothesized that: under the stress condition of LCEO, a cell wall stimulus, the amount of histidine in *A. baumannii* 17 is increased as for the cell membrane and two-component system reparation.

In bacteria, peptidoglycan is one of the building blocks of the cell wall [56]. Here, as shown in Figure 11B, the gene encoding an enzyme (UDP-N-acetylmuramoylalanine--D-glutamate ligase [EC:6.3.2.9]) that catalyzes the irreversible reaction in the early stages of peptidoglycan biosynthesis pathway was up-regulated. A recent report showed that LCEO caused damages to cell membranes of *A. baumannii* [27]. Therefore; with some related results above, we think that the cell wall of *A. baumannii* 17 was damaged by the LCEO, so that the bacteria required more peptidoglycan to do the repair work.

### 3.13. Effects of LCEO on Biofilm Formation and FICI Values

Studies have shown that LCEO affected the biofilm of *Borrelia burgdorferi* and *Candida* spp. [54,56]. Therefore, we investigated the effects of different concentrations of LCEO on the biofilm formation of *A. baumannii* 17. As shown in Figure 10B, the formation of biofilm of *A. baumannii* 17 was significantly promoted by the extremely low concentration of LCEO (1/4096 to 1/512 MIC). This result was similar to that of Junyan’s report: low level of ampicillin stimulation can accelerate the biofilm formation of *S. aureus* [57]. However, the biofilm biosynthesis of *A. baumannii* 17 was significantly reduced by the relatively high concentrations of LCEO (1/64 to 1/2 MIC). Compared to the planktonic bacteria, bacteria in biofilm are more resistant to antibiotics or reactive molecules produced by the host immune system [58]. Thus, at the right concentration, LCEO may be able to promote the efficacy of other combined antimicrobial drug(s).

Therefore, the FICI value was used to evaluate the synergistic growth inhibitory effect of LCEO with antibiotics. Here, we randomly selected some antibiotics (tetracycline, meropenem, chloramphenicol, amikacin, ampicillin, kanamycin, ciprofloxacin, fosfomycin, streptomycin sulfate, ceftiofur) for testing the synergistic growth inhibitory effect with LCEO. As shown in Table 4, synergy and additivity were found when LCEO combined with chloramphenicol and tetracycline on *A. baumannii* 17, and no antagonistic effect was recorded. This suggested that LCEO has certain potential research value on pathogenic microbial infection.

### 3.14. RT-PCR Validation

The qRT-PCR results of eight DEGs were very similar to those of the transcriptome analysis (Figure 12), indicating that the transcriptome analysis in this work was highly accurate and reliable.

## 4. Conclusions

Evidence of transcriptomic assays for the stress response of *A. baumannii* under low concentration of LCEO were reported in the present work; 320/352 DEGs were significantly up/down-regulated in LCEO-treated *A. baumannii*; 127 DEGs were significantly enriched in three GO terms (oxidation-reduction process; oxidoreductase activity; oxidoreductase activity, acting on the CH-CH group of donors), which indicated that the cell redox state was widely affected by 1/2 MIC of LCEO. LCEO may also inhibit the aerobic respiration, synthesis of ketone bodies and metabolism of some amino acids as well as promote fatty acid degradation of *A. baumannii*. Therefore, we hypothesized that LCEO, due to its inherent reductivity, extensively affected the redox reaction of cells, thus causing metabolic disorders of bacteria then exerting bacteriostatic effects. The integrity of the cell membrane of *A. baumannii* was damaged by sub-inhibitory concentration of LCEO. We also founded that the biofilm formation of *A. baumannii* was promoted/inhibited by extremely low/relatively high concentration of LCEO. LCEO and chloramphenicol have a synergistic growth inhibitory effect against *A. baumannii*. These results provided evidence that LCEO affects *A. baumannii* in multiple ways (multiple targets), and expand the knowledge of the stress response of *A. baumannii* under low concentration of LCEO.

## Figures and Tables

**Figure 1 genes-12-01003-f001:**
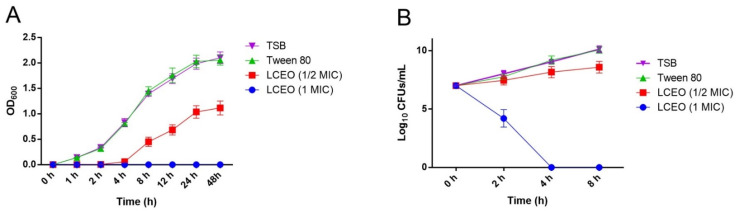
Growth curve (**A**) and time kill curve (**B**) assay of LCEO on *A. baumannii* 17. In tween 80 and LCEO (1 MIC) group, they had the same concentration (1.08 mg/mL) of tween 80.

**Figure 2 genes-12-01003-f002:**
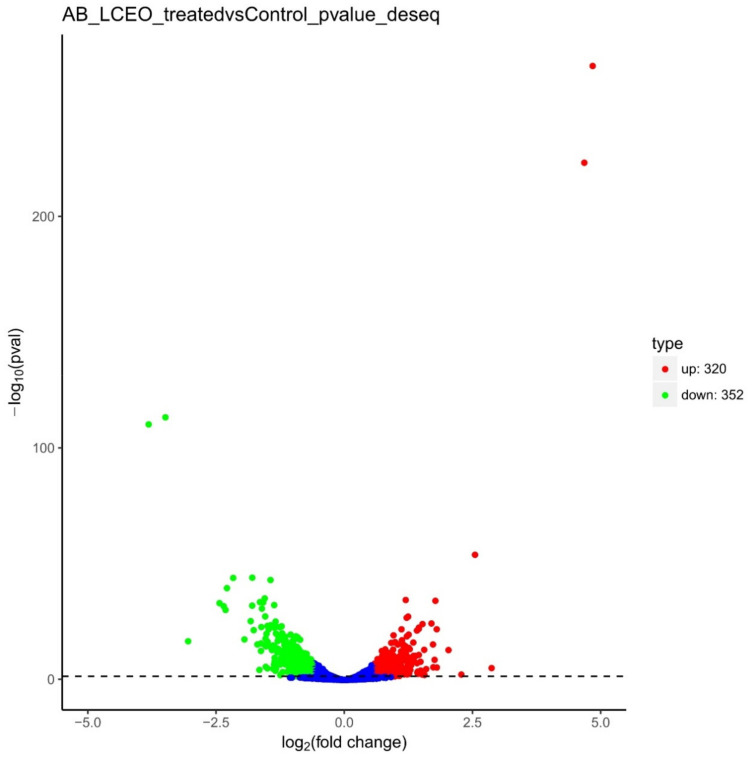
Differential expression level of LCEO-treated (AB_LCEO_treated) and non-treated *A. baumannii*(Control) groups identified by |fold change| > 1.5 and *p*-value < 0.05.

**Figure 3 genes-12-01003-f003:**
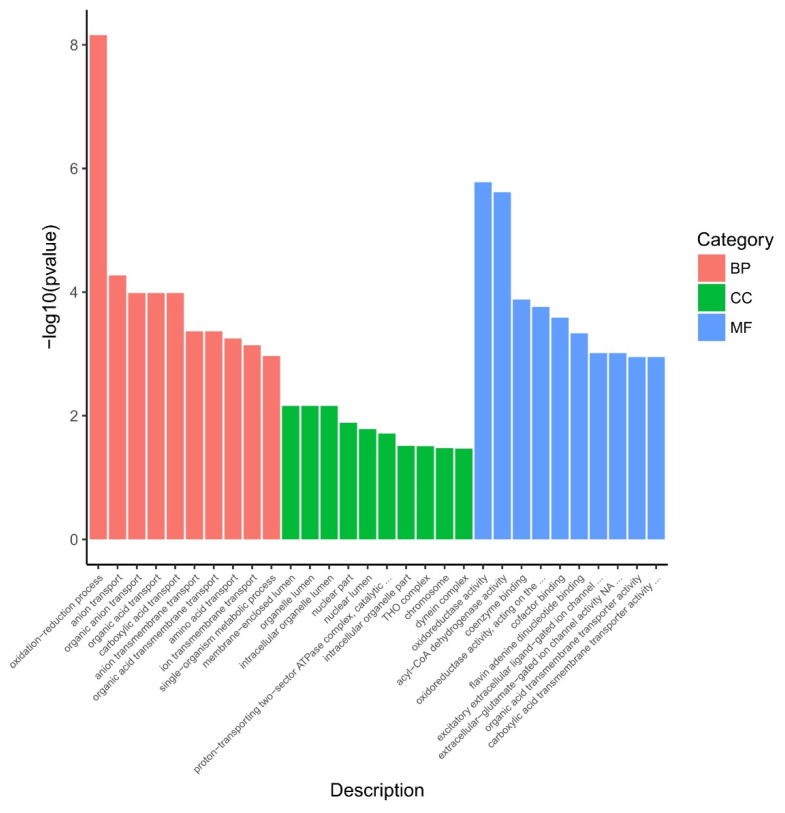
Significantly enriched Gene Ontology (GO) terms of differentially expressed genes. BP: biological process, CC: cellular component, MF: molecular function.

**Figure 4 genes-12-01003-f004:**
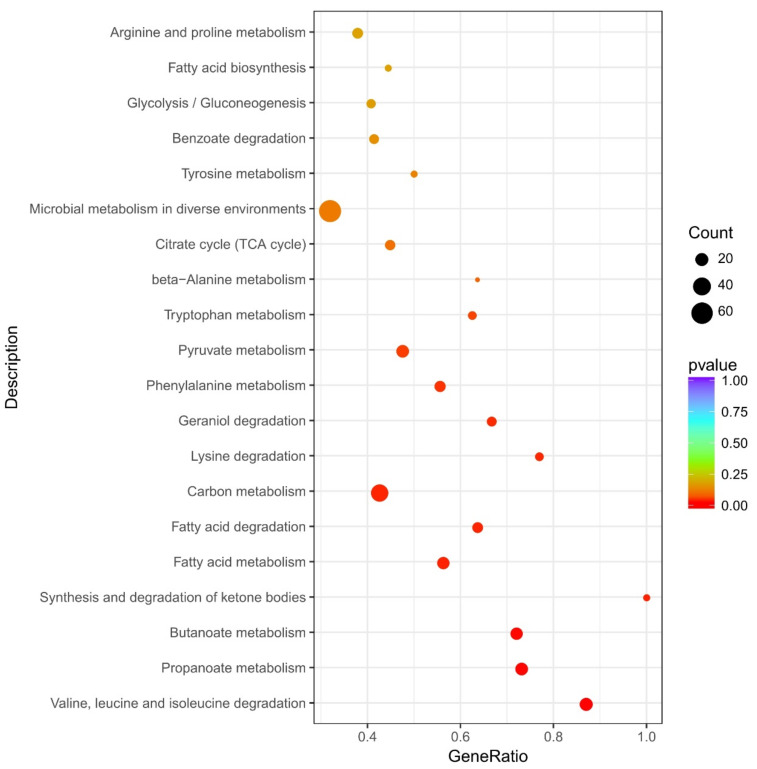
Significantly enriched Kyoto Encyclopedia of Genes and Genomes (KEGG) pathways of differentially expressed genes.

**Figure 5 genes-12-01003-f005:**
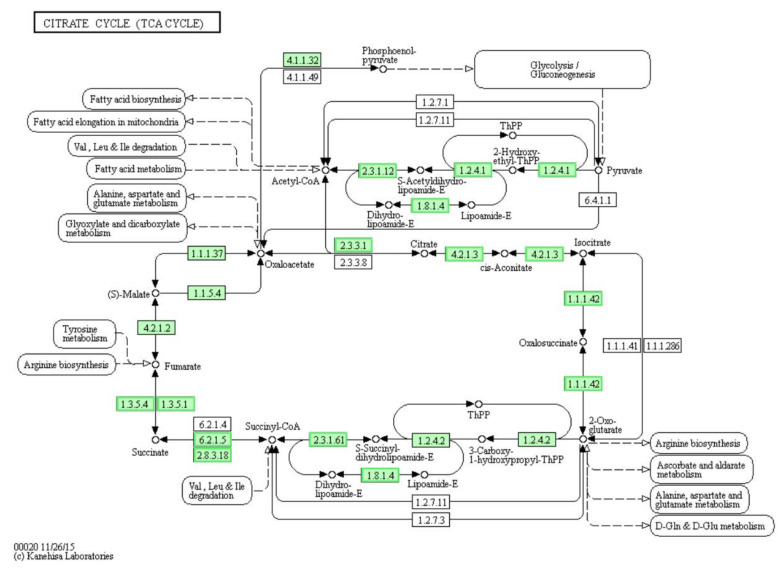
Significantly enriched KEGG pathway “Citrate cycle (TCA cycle)”; (from KEGG database, green rectangles represent proved organism-specific gene product and green frames represent down-regulation and red frames represent up-regulation).

**Figure 6 genes-12-01003-f006:**
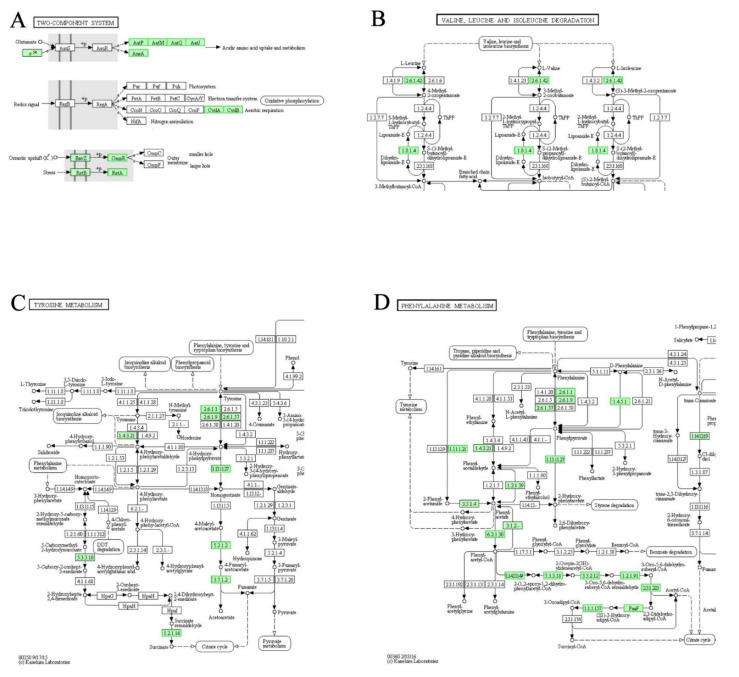
Significantly enriched KEGG pathway “Two-component system” (**A**), “Valine, leucine and isoleucine degradation” (**B**), “Tyrosine metabolism” (**C**) and “Phenylalanine metabolism” (**D**); (from KEGG database, green rectangles represent proved Organism-specific gene product and green frames represent down-regulation and red frames represent up-regulation).

**Figure 7 genes-12-01003-f007:**
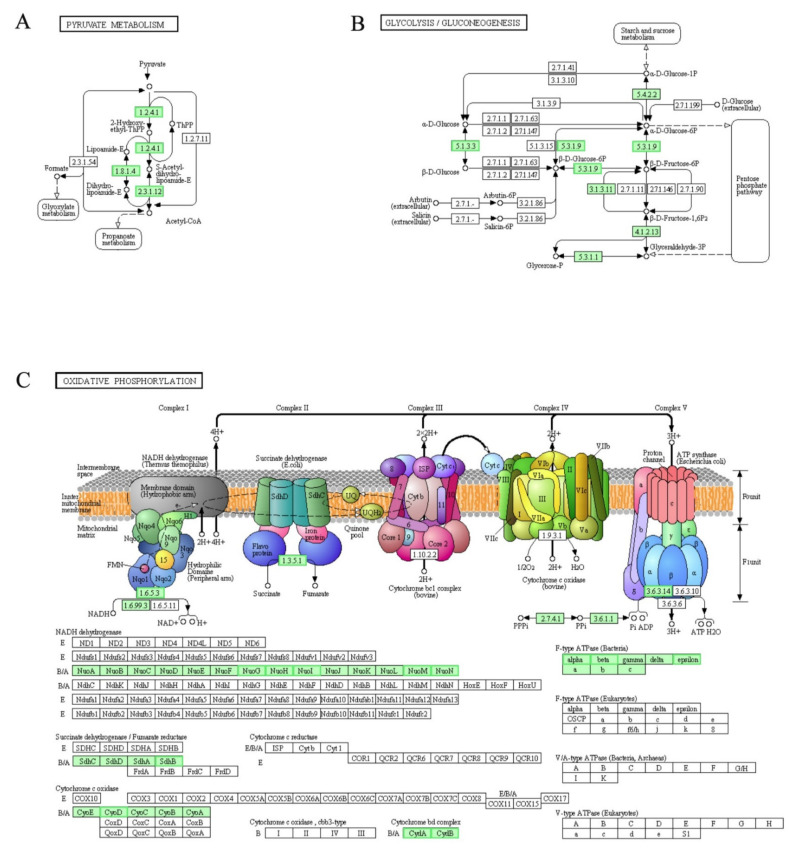
Significantly enriched KEGG pathway “Pyruvate metabolism” (**A**), “Glycolysis / Gluconeogenesis” (**B**) and “Oxidative phosphorylation” (**C**); (from KEGG database, green rectangles represent proved Organism-specific gene product and green frames represent down-regulation and red frames represent up-regulation).

**Figure 8 genes-12-01003-f008:**
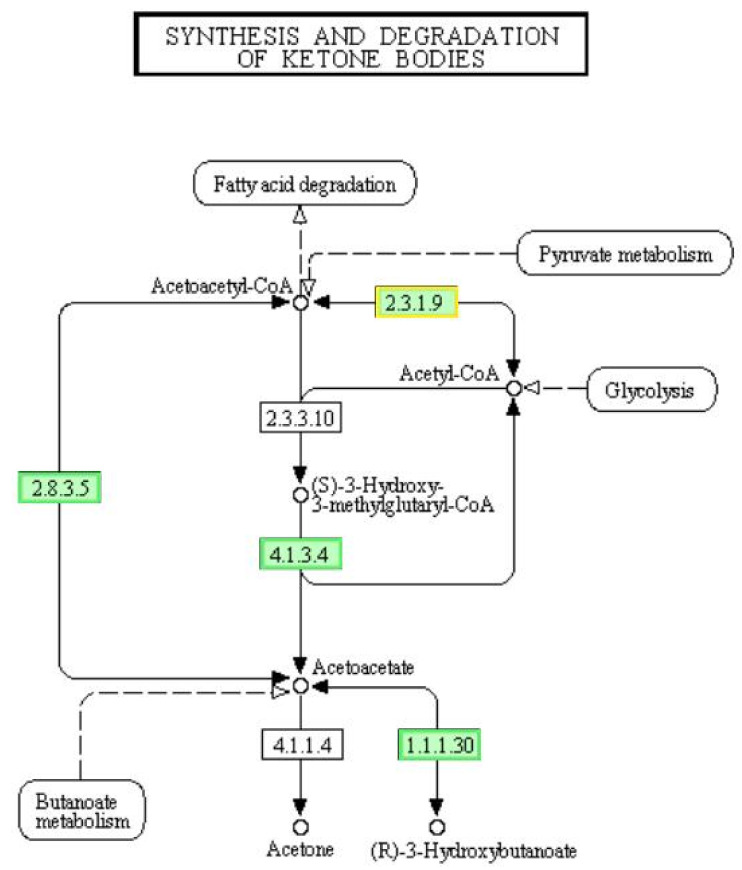
Significantly enriched KEGG pathway “Synthesis and degradation of ketone bodies”; (from KEGG database, green rectangles represent proved Organism-specific gene product and green frames represent down-regulation and red frames represent up-regulation).

**Figure 9 genes-12-01003-f009:**
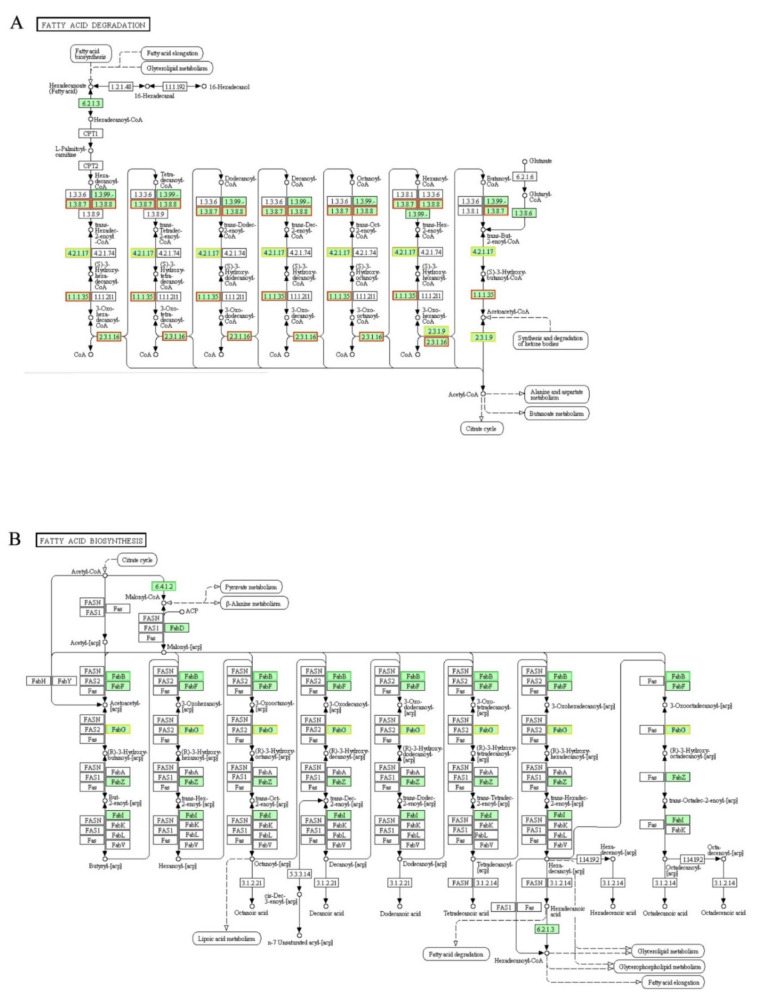
Significantly enriched KEGG pathway “Fatty acid degradation” (**A**) and “Fatty acid biosynthesis” (**B**); (from KEGG database, green rectangles represent proved Organism-specific gene product and green frames represent down-regulation and red frames represent up-regulation).

**Figure 10 genes-12-01003-f010:**
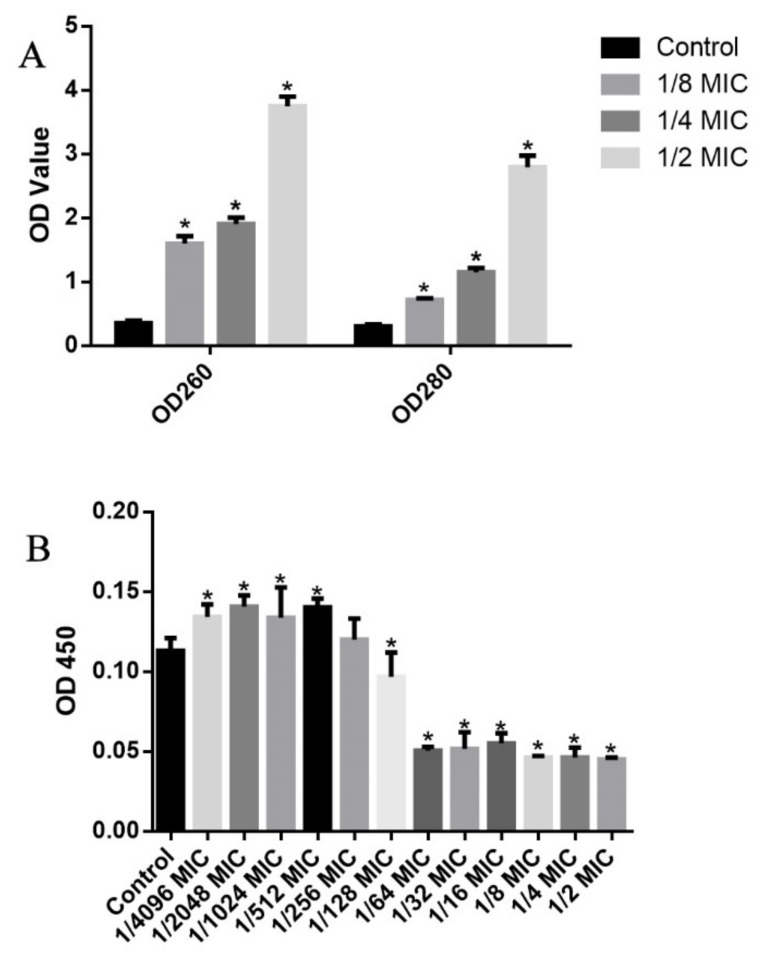
Effects of different concentrations of LCEO on intracellular DNA and protein leakages (**A**) and biofilm biosynthesis (**B**) of *A. baumannii* 17. OD_260_, OD_280_ and OD_450_ value represent the levels of intracellular DNA leakages, intracellular protein leakages and biofilm biosynthesis in different groups, respectively. Note: The data represent the means ± standard deviation of the results. The “*” on the top of a data column indicates a significant difference between this group and the control group (*p* < 0.05).

**Figure 11 genes-12-01003-f011:**
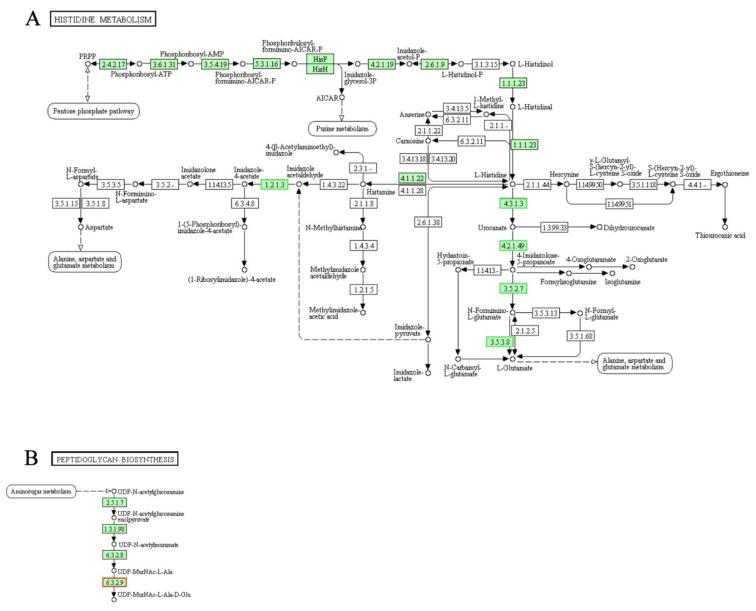
Significantly enriched KEGG pathway “Histidine metabolism” (**A**) and “Peptidoglycan biosynthesis” (**B**); (from KEGG database, green rectangles represent proved Organism-specific gene product and green frames represent down-regulation and red frames represent up-regulation, yellow frames represent both down and up-regulation).

**Figure 12 genes-12-01003-f012:**
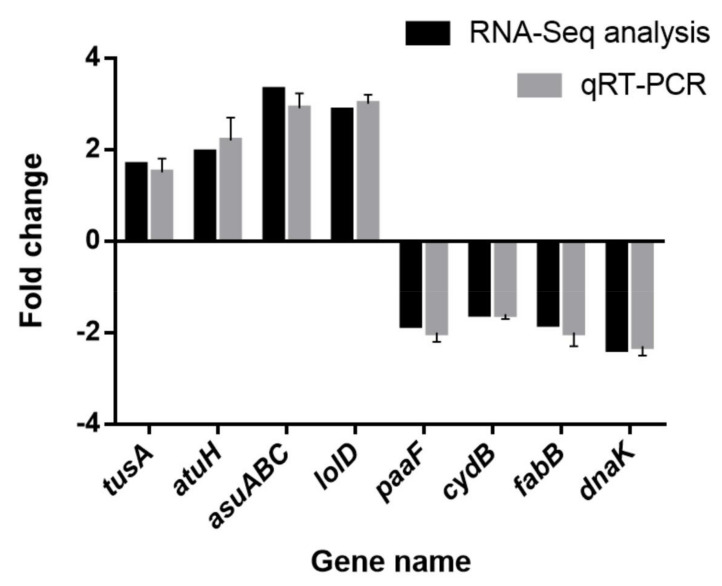
Validation of RNA sequencing data by quantitative real-time PCR. Note: The data represent the means ± standard deviation of the results.

**Table 1 genes-12-01003-t001:** Chemical compositions of *L. cubeba* essential oil (LCEO).

Composition	Ratio (%)
α-Citral	38.28
β-Citral	29.29
Cinene	16.5348
Eudesmol	2.56
Citronellal	2.2383
Trans-Verbenol	2.06
Aromatic alcohol	1.86
β-Pinene	1.5104
Cis-Verbenol	1.3771
α-Vinyl acetate	1.0945
Geraniol	0.93
α-Pinene	0.7307
2-Methyl-2-Hexen-6-One	0.6994

**Table 2 genes-12-01003-t002:** Primer sequences used for quantitative real-time polymerase chain reaction (RT-PCR).

Gene Name	Primer Sequence (5′->3′)	NCBI-Protein ID	Products (Base Pairs)
*tusA*	AGATGTTGTTGAAGTGTT CCAGATAACGGTATTCTT	ABO10836	137
*atuH*	AATGATTGGTAATGGTATG AACTGGATAATGGCATAT	ABO10859	174
*asuABC*	GGAGTGATGATGCTAGGT TAGAATAGGAAGGATGCCATA	ABO10525	96
*lolD*	AGCAGACAGTCAATATAAGT GGTCCATACGATGAGATAG	ABO13028	80
*paaF*	AAGAAGAATTACTTGAACA TATTGAATGGAACATACG	ABO10593	150
*cydB*	AATATCATTCCACCATCA ATAACCTTCATCACCTAC	ABO11861	165
*fabB*	GGAACGAGTTGTCATCAC GTTATAGCGAATGCCAGAG	ABO10573	106
*dnaK*	TCTACTGCTGCTGATAAC GATGTCACCTAACTGGAA	ABO13363	108

**Table 3 genes-12-01003-t003:** Minimum inhibitory concentration (MIC) values (mg/mL) of tested components on *A. baumannii* strains.

Component	*A. baumannii* 4	*A. baumannii* 20	*A. baumannii* 17
Ceftriaxone Sodium	3.2	1.6	0.8
Amoxicillin	0.8	0.8	1.6
Berberine	0.0015625	0.003125	0.003125
Neomycin	0.4	0.2	3.2
Levofloxacin	0.0125	0.025	0.025
Fosfomycin	3.2	3.2	0.4
Colistin	0.0015625	0.0015625	0.0015625
Mequindox	0.1	0.2	0.1
Clinfloxacin	0.025	0.025	0.025
Sulfamonomethoxine	6.4	6.4	3.2
Gatifloxacin	0.003125	0.025	0.025
Amikacin	0.8	6.4	3.2
Ceftazidime	0.2	0.025	0.1
Lincomycin	0.4	0.4	3.2
Ceftiofur Sodium	3.2	0.8	0.4
Florfenicol	0.32	0.16	0.32
Azithromycin	0.8	0.4	3.2
Cefotaxime Sodium	0.0125	0.2	0.1
Rifampin	0.0125	0.05	0.025
Meropenem	0.2	0.1	0.2
Ceftriaxone Sodium	3.2	1.6	0.8
LCEO	1.08	1.08	1.08
Tween 80	>108	>108	>108

**Table 4 genes-12-01003-t004:** Fractional Inhibitory Concentration Index (FICI) values of LCEO with some antibiotics on *A. baumannii* 17.

Drugs	MIC Alone	MIC Combined	FICI	Interpretation
LCEO	1.08 mg/mL	0.27 mg/mL	0.75	ADD
Tetracycline	1.28 mg/mL	0.64 mg/mL		
LCEO	1.08 mg/mL	0.27 mg/mL	1.25	IND
Meropenem	0.32 mg/mL	0.32 mg/mL		
LCEO	1.08 mg/mL	0.27 mg/mL	0.375	SYN
Chloramphenicol	0.32 mg/mL	0.04 mg/mL		
LCEO	1.08 mg/mL	0.27 mg/mL	1.25	IND
Amikacin	1.28 mg/mL	1.28 mg/mL		
LCEO	1.08 mg/mL	1.08 mg/mL	2	IND
Ampicillin	0.32 mg/mL	0.32 mg/mL		
LCEO	1.08 mg/mL	1.08 mg/mL	2	IND
Kanamycin	0.32 mg/mL	0.32 mg/mL		
LCEO	1.08 mg/mL	0.27 mg/mL	1.25	IND
Ciprofloxacin	0.64 mg/mL	0.64 mg/mL		
LCEO	1.08 mg/mL	0.54 mg/mL	1.5	IND
Fosfomycin	0.64 mg/mL	0.64 mg/mL		
LCEO	1.08 mg/mL	1.08 mg/mL	2	IND
Streptomycin sulfate	1.28 mg/mL	1.28 mg/mL		
LCEO	1.08 mg/mL	0.54 mg/mL	2.5	IND
Ceftiofur	0.64 mg/mL	1.28 mg/mL		

Notes: Relationship between FICI values and drug combinations: synergy (SYN): <0.5; additivity (ADD): 0.5–1; indifference (IND): >1–4; and antagonism (ANT): >4.

## Data Availability

All datasets generated for this study are included in the article/Supplementary Material. This manuscript and data have not been submitted for possible publication to another journal or that the work has previously been published elsewhere. The raw sequence data reported in this paper have been deposited in the Genome Sequence Archive (Genomics, Proteomics and Bioinformatics 2017) in BIG Data Center (Nucleic Acids Res 2019), Beijing Institute of Genomics (BIG), Chinese Academy of Sciences, under accession numbers CRA003099 that is publicly accessible at https://bigd.big.ac.cn/gsa (accessed on 18 August 2020).

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
