# Peer review of "Transcriptomic Analysis of Drug-Resistance Acinetobacter baumannii under the Stress Condition Caused by Litsea cubeba L. Essential Oil via RNA Sequencing"

_genes, 2021, doi:10.3390/genes12071003_

Round 1

Reviewer 1 Report

Abstract:

I have read the manuscript “Transcriptomic analysis of Drug-Resistance Acinetobacter baumannii under the stress condition caused by Litsea cubeba L. essential oil via RNA sequencing”. Below is my input for your perusal:

Abstract:

Lines 11-13: Change sentence to:However, research on stress response of drug-resistant A. baumannii under sub-lethal LCEO concentrations had been limited so far.”

Line 15: Change to: up or down-regulated, respectively, in LCEO-treated A. baumannii.”

Line 16: What are GO terms? Please clarify in manuscript.

Line 19: What is a KEGG enrichment? Please clarify in manuscript.

Line 23: What is a FICI value? Please clarify or omit.

Lines 23-25: Change sentence to: “Our results indicate that (the growth of?) A. baumannii may be inhibited by LCEO, and give insights into the stress response of A. baumannii under sub-lethal concentrations of LCEO.

Introduction

Line 38: What is meant by “in present”?

Line 43: Change to: “drug-resistant bacteria” (bacteria is the plural of bacterium).

Lines 47-48: Change to: can cure infections and prevent outbreaks caused by A. baumannii, as this pathogen has multi drug resistant properties”.

Lines 52-53: Change to: “Therefore, the elucidation of the mechanisms by which LCEO impacts A. baumannii is helpful for the development of ways to use LCEO.”

Lines 56-57: Change to: “The objective of the present work was to detect stress reactions of MDR A. baumannii under ½ MIC of LCEO using transcriptomics.”

Materials and methods:

Line 80: Change recruited to performed.

Line 120: Change to “The method applied was a slight modification of the method used in a previous work.”

Results and discussion

I recommend a major revision of the results and discussion. The results need to be displayed more clearly in comparison with the controls and brought into context of existing literature. As it is, the discussion looks more like a textbook chapter to me than the discussion of results obtained.

Author Response

Abstract:

Lines 11-13: Change sentence to: “However, research on stress response of drug-resistant A. baumannii under sub-lethal LCEO concentrations had been limited so far.”

--Done.

Line 15: Change to: “up or down-regulated, respectively, in LCEO-treated A. baumannii.”

--Done.

Line 16: What are GO terms? Please clarify in manuscript.

--An explanation has been added at the beginning of Section 3.5.

Line 19: What is a KEGG enrichment? Please clarify in manuscript.

--An explanation has been added at the beginning of Section 3.6.

Line 23: What is a FICI value? Please clarify or omit.

--FICI value was explained in Section 2.9.

Lines 23-25: Change sentence to: “Our results indicate that (the growth of?) A. baumannii may be inhibited by LCEO, and give insights into the stress response of A. baumannii under sub-lethal concentrations of LCEO.

--Done.

Introduction

Line 38: What is meant by “in present”?

--Done.

Line 43: Change to: “drug-resistant bacteria” (bacteria is the plural of bacterium).

--Done.

Lines 47-48: Change to: “can cure infections and prevent outbreaks caused by A. baumannii, as this pathogen has multi drug resistant properties”.

--Done.

Lines 52-53: Change to: “Therefore, the elucidation of the mechanisms by which LCEO impacts A. baumannii is helpful for the development of ways to use LCEO.”

--Done.

Lines 56-57: Change to: “The objective of the present work was to detect stress reactions of MDR A. baumannii under ½ MIC of LCEO using transcriptomics.”

--Done.

Materials and methods:

Line 80: Change recruited to performed.

--Done.

Line 120: Change to “The method applied was a slight modification of the method used in a previous work.”

--Done.

Results and discussion

I recommend a major revision of the results and discussion. The results need to be displayed more clearly in comparison with the controls and brought into context of existing literature. As it is, the discussion looks more like a textbook chapter to me than the discussion of results obtained.

 --Currently, there are not many annotations about gene product of A. baumannii in the KEGG database, so many results cannot be correlated very well like other RNA-seq paper. Of course, I have tried to find relationships between the pathways that these DEGs were enriched in. There were some findings, and they were listed. In every sub section of the results and discussion section, I followed four steps when I was writing: (1) introduce the pathways (related backgrounds); (2) present current results; (3) compare current results with the results of past similar studies; (4) make proper discussions and hypotheses. I have refined the pathways introducing at this time. Of course, other reviewers also put specific requirements for the revision of this section, and I revised them carefully according to the requirements. I was referring to this paper (https://doi.org/10.3389/fmicb.2018.02413) when I wrote the manuscript, and he also expressed in this way.

Reviewer 2 Report

3 major points:

  1. Transcriptomic analyses of the control should be assessed with the same amount of tween-80 as treated cells. Otherwise, we cannot rule out that the different gene regulation is due to this detergent rather than the LCEO. So I suggest to check this point by performing some analyses with A. baumannii grown in TSB, TSB + Tween-80, and TSB + LCEO.
  2. If you address the above point, please revise the rationale of the experimental design and the flow of the paragraphs which, at the moment, is very difficult to follow. Especially the conclusion of each paragraph is missing. In addition, please shorten it substantially.
  3. The final conclusions are not "conclusive". You addressed several metabolic pathways but did not delineate a mechanism by which LCEO acts on the physiology of A. baumannii cells. It is reasonable to hypthesize an involvement, but the authors did not demonstrate a true LCEO involvement.

In addition, I'd suggest to add the link of  each KEGG pathway instead of showing parts of it in a figure.

Author Response

Transcriptomic analyses of the control should be assessed with the same amount of tween-80 as treated cells. Otherwise, we cannot rule out that the different gene regulation is due to this detergent rather than the LCEO. So I suggest to check this point by performing some analyses with A. baumannii grown in TSB, TSB + Tween-80, and TSB + LCEO.

-- Thank you for your mentions. This is really necessary, we had certain oversights. We appended some experiments immediately. The MIC value of tween 80 on A. baumannii was more than 100 mg/mL. Also, the results of growth curve and time-kill curve suggested that the concentration (1.08 mg/mL) of tween 80 which used in the corresponding experiment did not affect the growth of A. baumannii. Therefore, we believed that in this research, the different gene regulations were due to the LCEO rather than the tween 80. In other words, there is no need to perform transcriptomic analyze of A. baumannii grown in TSB + tween 80. Here are some references to the fact that tween 80 did not affect bacterial growth:

https://doi.org/10.1039/C2FO10198J

https://doi.org/10.1016/j.ultsonch.2016.10.020

If you address the above point, please revise the rationale of the experimental design and the flow of the paragraphs which, at the moment, is very difficult to follow. Especially the conclusion of each paragraph is missing. In addition, please shorten it substantially.

--I added experimental design and data so it should make sense. I have a short summary at the end of every paragraph. Some of the descriptions that were too verbose or inappropriate have been revised.

The final conclusions are not "conclusive". You addressed several metabolic pathways but did not delineate a mechanism by which LCEO acts on the physiology of A. baumannii cells. It is reasonable to hypothesize an involvement, but the authors did not demonstrate a true LCEO involvement.

--Done.

Reviewer 3 Report

This study investigates the response of A. baumannii---and important opportunistic human pathogen--to essential oil stress. The authors seek to determine the response of this organism to sub-MIC concentrations by looking at the transcriptome in a first-step study to investigate mechanistic response to essential oils and allied chemistries.

While this reviewer recommends the publishing of this work it is worthy to note a few comments:

It was not clear to this reviewer why the authors chose the genes in Table 2 in their approach to validate RNA seq data. The statistical approach using ANOVA seems rigorous enough, but what about the number of targets assessed? Is 8 enough to satisfy the authors of the rigor of their seq data?

Further discussion of the significance of redox altering drugs on the pathogenic life history of A. baumannii would lend the paper more weight. How do drugs that influence membrane biology, PMF and metabolic flux affect the overall drug resistance and infectious phenotype of this pathogen--especially in light of the large volume of published work that exists on this topic in pathogens such as S. aureus?

As with all transcriptome studies it suffers from being a bit descriptive, w/o seeking to determine underlying mechanism. This is not a cheap-shot, as most do and the data is valuable. It may be of worth to elaborate on future approach to uncover mechanisms that determine these transcriptomic alterations.

Author Response

It was not clear to this reviewer why the authors chose the genes in Table 2 in their approach to validate RNA seq data. The statistical approach using ANOVA seems rigorous enough, but what about the number of targets assessed? Is 8 enough to satisfy the authors of the rigor of their seq data?

--The reasons why these 8 genes were selected were mentioned in the materials and methods section: To validate the RNA-seq results, 8 DEGs were selected for quantitative real-time PCR (RT-PCR) assay. The selective criteria of these 8 DEGs were: (1) in different KEGG pathways; (2) the |fold change| > 1.5; (3) 4 DEGs were up-regulated and the remaining 4 DEGs were down-regulated.

8 genes are sufficient. And now, the error rate of transcriptome technology is very low. Here are some references, and the last one was published in this journal.

https://doi.org/10.1016/j.foodcont.2018.08.021

https://doi.org/10.3389/fmicb.2018.02413

https://doi.org/10.3390/genes12040536

Further discussion of the significance of redox altering drugs on the pathogenic life history of A. baumannii would lend the paper more weight. How do drugs that influence membrane biology, PMF and metabolic flux affect the overall drug resistance and infectious phenotype of this pathogen--especially in light of the large volume of published work that exists on this topic in pathogens such as S. aureus?

-- We believe that in order to discuss the effect of LCEO on the REDOX states of A. baumannii, more phenotypic tests are needed. And there are still gapes for further research in areas such as REDOX. However, due to the experimental funds and other reasons, we plan to perform the in-depth work of this subject in the future.

As with all transcriptome studies it suffers from being a bit descriptive, w/o seeking to determine underlying mechanism. This is not a cheap-shot, as most do and the data is valuable. It may be of worth to elaborate on future approach to uncover mechanisms that determine these transcriptomic alterations.

-- As mentioned above, more phenotypic tests may be needed to clarify these issues. On the other hand, there are not many annotations about gene product of A. baumannii in the KEGG database currently. So, many results cannot be correlated very well like other RNA-seq paper. Of course, I have tried to find relationships between the pathways that these DEGs were enriched in. There were some findings, and they were listed. At this moment, maybe that is the best we did. Everyone can download the raw data of RNA-seq in this work as it was uploaded to a public database. Therefore, when there are more annotations in the future and/or if someone needs this data, it can be used directly again.

Round 2

Reviewer 2 Report

The manuscript was improved.

However, there are several grammatical and mistyping mistakes.

I do believe that a link to the kegg website would be enough to explain the results instead of those pictures.
